# Fish Consumption: Influence of Knowledge, Product Information, and Satisfaction with Product Attributes

**DOI:** 10.3390/nu14132691

**Published:** 2022-06-28

**Authors:** Greta Krešić, Elena Dujmić, Dina Lončarić, Snježana Zrnčić, Nikolina Liović, Jelka Pleadin

**Affiliations:** 1Department of Food and Nutrition, Faculty of Tourism and Hospitality Management, University of Rijeka, Primorska 46, P.O. Box 97, 51410 Opatija, Croatia; gretak@fthm.hr (G.K.); nikolina.liovic@fthm.hr (N.L.); 2Center for Projects, Faculty of Tourism and Hospitality Management, University of Rijeka, Primorska 46, P.O. Box 97, 51410 Opatija, Croatia; elena.dujmic@fthm.hr; 3Department of Marketing, Faculty of Tourism and Hospitality Management, University of Rijeka, Primorska 46, P.O. Box 97, 51410 Opatija, Croatia; dinal@fthm.hr; 4Laboratory for Fish Pathology, Croatian Veterinary Institute, Savska Cesta 143, 10000 Zagreb, Croatia; zrncic@veinst.hr; 5Laboratory for Analytical Chemistry, Croatian Veterinary Institute, Savska Cesta 143, 10000 Zagreb, Croatia

**Keywords:** consumer behaviour, fish consumption, PLS-SEM, product information, product attributes, subjective knowledge

## Abstract

Due to its numerous health benefits, fish consumption should be strongly encouraged. Fish consumption, however, is a complex phenomenon influenced by various factors. The aim of this research is to examine the influence of knowledge, product information, and satisfaction with product attributes on fish consumption in a nationally representative sample of people responsible for food purchasing within households in Croatia (n = 977) and Italy (n = 967). Fish consumption was well predicted (R^2^ = 15%) by the proposed structural model, using the partial least squares structural equation modelling method (PLS-SEM). The obtained results confirm that subjective knowledge (β = 0.277, *p* < 0.001) and satisfaction with product attributes (β = 0.197, *p* < 0.001) are predictors of fish consumption. Subjective knowledge was influenced by product information (β = 0.161, *p* < 0.001), as well as by satisfaction with product attributes (β = 0.282, *p* < 0.001), while objective knowledge had an influence on product information (β = 0.194, *p* < 0.001). Although satisfaction with product attributes was the strongest predictor of subjective knowledge in both countries (β_CRO_ = 0.244, β_IT_ = 0.398), it had a greater effect among Italians (*p* = 0.001), while the impact of product information (β_CRO_ = 0.210, β_IT_ = 0.086) was more pronounced among Croatians (*p* = 0.010). Since the mediating role of subjective knowledge in all models was confirmed, action focused on enhancing subjective knowledge should be taken to increase fish consumption.

## 1. Introduction

An unhealthy diet, characterised by an excessive intake of energy-dense food and a lack of high-quality proteins and nutrient-dense food, is a growing problem in developing and developed countries, causing numerous negative health impacts.

Fish consumption can contribute to correcting high-calorie and low-nutrient-dense diets [1]. Fish, as a low-energy, high-protein food, could also contribute to the intake of some essential nutrients, such as iodine, calcium, selenium, and vitamins D and A. Of particular interest are n-3 long-chain polyunsaturated fatty acids (n-3 LCPUFA) due to their anti-inflammatory and cardioprotective effects [2]. Consequently, regular fish consumption could reduce the burden of non-communicable diseases associated with overweight or obesity, as well as micronutrient deficiencies, mortality from cardiovascular disease, heart failure, and stroke [3,4]. A systematic review and meta-analysis confirmed that n-3 PUFA intake via fish consumption is associated with a lower risk of depression [5].

In the light of the trend of including territorial diets in national food-based dietary guidelines, it is well accepted that the Mediterranean diet as a plant-based diet—characterised by an abundance of fruits, vegetables, fish, seeds, legumes, moderate amounts of dairy foods, and moderate to very little amounts of other animal-sourced proteins—lowers the incidence of chronic diseases and improves longevity [6].

National and international health authorities and policy makers have an interest in, as well as an obligation for, promoting fish consumption. Recently, extensive research has been conducted among policy documents sourced in the UN FAO and WHO databases, with the aim of investigating policy inclusion between fisheries and public health nutrition policies [7]. The results indicate that only 68 out of 165 public health nutrition policies identified the importance of fish consumption as a key objective. Additionally, just 77 out of 158 national fisheries policies identified nutrition as a key objective in the sector. Countries with higher overweight prevalence have public health nutrition and fisheries policies that are not aligned [7].

Croatia and Italy, although they are neighbouring Mediterranean countries sharing the Adriatic Sea, are complete opposites in terms of fish consumption and the health status of their citizens. As regards annual per capita fish consumption, Italy is among the top six EU countries (31.2 kg), while Croatia, with a consumption of 20.8 kg, is in 13th place out of the 27 countries [8]. Additionally, Croatia (alongside Malta) has the highest proportion of overweight and obese adults in its population, with 65% of adults belonging to these categories; meanwhile, Italy, with a share of 46%, has the lowest prevalence of overweight and obesity among adults in the EU [9]. Furthermore, deaths from cardiovascular diseases account for 42% of total deaths in Croatia, while in Italy, the share is somewhat lower (34%) [10]. When asked about their self-perceived health status, 73% of Italians identified their health as very good or good, whereas the same was the case for 60% of Croatians [11]. These two countries are worth comparing because, given Italy’s vicinity and similar culture and lifestyle habits, it could pose as a role model for increasing fish consumption in Croatia.

Fish consumption is complex and influenced by various factors. Many studies propose that knowledge plays an important role in explaining fish consumption behaviour [12,13,14,15,16]. In order to assess the influence of knowledge on consumers’ behaviour, a distinction should be made between subjective and objective knowledge. Objective knowledge refers to a consumer’s factual knowledge, while subjective knowledge can be thought of as the consumer’s perception of how much he/she knows about the product [17].

Furthermore, research also suggests a link between fish consumption, knowledge, and usage of information cues [12,14,16].

Greater involvement in product information search and selection is observed among consumers with a higher level of objective knowledge [12,18]. This group of consumers also has a better understanding of the meaning of product information and is less influenced by marketing misinformation [18,19]. Additionally, a high interest in and usage of information cues is one of the characteristics of consumers with the highest levels of fish consumption, who also exhibit the highest subjective and objective knowledge about fish [14,16].

Satisfaction with product attributes may play an important role in future fish consumption. Customer satisfaction is an individual’s perception of the performance of the product or service relative to their expectations [20]. Consumer experiences and post-purchase evaluations play a major role in future purchase decisions [21]. Previous research has confirmed that consumer satisfaction influences intention to repurchase food [22]. This has also been confirmed in research focused on fishery products. For example, satisfaction was confirmed to significantly affect repurchase loyalty [23]. In addition, it has been shown that consumers who are satisfied with safety are more likely to consume live fish more frequently [24]. Therefore, we can assume that satisfaction with a purchase or individual product attributes can be expected to determine future purchases. Furthermore, a recent study investigated the relationship between customer satisfaction, loyalty, knowledge, and competitiveness in the food industry [25]. The study confirmed that increased loyalty leads to increased knowledge, and that the rate of repeat purchases of a product is important for the relationship between the variables. Better-informed customers can be more loyal to the product, which leads to competitive advantage and profit [25]. Thus, in our research, customer satisfaction with product attributes focuses on cumulative satisfaction, which can be understood as long-term and is based on repeated purchases and consumers’ overall experience with a product. Therefore, consumers who are satisfied with product attributes accumulate the experience and improve their knowledge, which has a positive impact on future fish consumption.

Fish consumption habits, attitudes, and preferences in Italy and Croatia have been previously studied, focusing on several topics, including fresh fish, the preference for wild and farmed fish, the influence of health attitudes and sociodemographic characteristics, and perceptions of different types of fish products [26,27,28,29]. However, to the best of our knowledge, none of these studies have examined the influence of knowledge on fish consumption. 

Based on the consensus that knowledge is a key construct in information acquisition, the evaluation of product attributes and, consequently, in the decision-making process—but also taking into account the fact that there is little literature in Croatia and Italy—the aim of this work is to examine the influence of knowledge, the importance of product information, and satisfaction with product attributes on fish consumption.

In this regard, partial least squares structural equation modelling (PLS-SEM) was applied. PLS-SEM is a causal–predictive approach that allows the testing of a theoretical framework [30] in which, for the purpose of this study, subjective knowledge had a mediating role between fish consumption and other examined variables. Until now, no research has applied this method in the proposed context. Earlier on, PLS-SEM was used to determine factors influencing fresh fish consumption using an expanded Theory of Planned Behaviour [28], while the study that investigated the role of health-related beliefs and consumer knowledge on fish consumption used the covariance-based SEM (CB-SEM) method [15]. Therefore, our study is unique in its nature and methodological approach.

## 2. Materials and Methods

### 2.1. Data Collection

The data collection fieldwork was performed in December 2019, by the professional market research agency Ipsos among its consumer panel members, using the CAWI (Computer-Aided Web Interviewing) method.

Stratified random sampling and proportional quota in line with national population distribution for gender, age and region were performed, according to the latest estimates of the State Statistical Offices of Croatia and Italy. Participants were a nationally representative sample of people who are responsible for food purchasing within the household in Croatia and Italy. The initial sample size consisted of 1000 respondents, aged 18 to 65, from each country

The final eligible study sample that could be considered as fishery product consumers was selected based on a positive response to the question: “Did you consume fishery products in the last 12 months?” The final study sample consisted of 977 participants in Croatia and 967 participants in Italy.

### 2.2. Questionnaire

For the purpose of this research, data were extracted from a large survey conducted among Croatian and Italian fishery consumers, and included the following parts: socio-demographic characteristics, frequency of fish consumption, objective knowledge, subjective knowledge, importance of product information, and satisfaction with product attributes.

Data presented in this paper are part of the extended research conducted within the framework of the European project AdriAquaNet (AdriAquaNet, Interreg V-A Italy–Croatia 2014–2020 Program, Blue innovation, ID10045161, Grant Agreement No. 36008). The research instrument used for the purpose of this study is available in Appendix A.

Fish consumption was measured as the self-reported frequency of consumption of two generic categories of fish: white fish and fatty fish. This classification comes from their differences in nutritional composition, price, and sensory properties, which could be important for consumers, and is common in several research papers [26,31,32,33]. Moreover, the European Food-Based Dietary Guidelines additionally emphasise the specificity of the consumption of fatty fish [2]. Levels of consumption frequency were encoded in seven ranked ordinal categories: once a year or less, once in 3 months, 2–3 times a month, once a week, 2–3 times a week, 4–5 times a week, or every day. Global single-item was calculated as sum of white and fatty fish consumption recoded as frequencies per year and aggregated to compute one variable.

Consumers’ level of objective knowledge was measured using 7 statements that were either true or false, based on previous studies [12,16,34]. Two of the statements were false—“Fish is a source of dietary fibre” and “The sea bass and sea bream available in the European market are exclusively wild species”—while the rest were true. A true/false scale was used, without including the “do not know” category in order to force participants to make up their minds about the proposed statements [34]. The number of correct answers was summed for each participant, giving an aggregated score of objective knowledge on a scale of 0–7 [16]. Subjective knowledge was measured using 4 statements on a 5-point Likert scale, ranging from 1 (strongly disagree) to 5 (strongly agree), based on a previous study [35].

The level of satisfaction with product attributes was also measured on a 5-point Likert scale, where 1 indicated extremely dissatisfied and 5 extremely satisfied. Six items were included in this construct, which are believed to be the main product attributes that consumers evaluate when purchasing fish: quality, freshness, choice, availability, price–quality ratio, and price. Results are expressed as the percentage of participants who rated their level of satisfaction with scores of 4 and 5 on the Likert scale (top 2 boxes).

The importance of product information included 11 possible information cues: shelf life, production method, country of origin, previous freezing, processing method, quality label, list of ingredients, eco-label, nutritional value, brand, and recommended method of preparation. The majority were mandatory cues for either unpackaged or packaged products, while quality label and eco-label were voluntary cues, chosen because of their great rise in importance among consumers. A 5-point Likert scale was used, ranging from 1 (not at all important) to 5 (very important). Results are expressed as the percentage of participants who rated their level of satisfaction with scores of 4 and 5 on the Likert scale (top 2 boxes).

The questionnaire was developed in English and further translated into Croatian and Italian by professional translators. The back-translation method was performed to ensure the quality and accuracy of the translation.

### 2.3. Statistical Analysis

Descriptive statistics were used to describe the sample and items included in the measurement scale. Statistical significance testing between the two countries was computed using an independent t-test and Pearson’s chi-squared test. Data were analysed using the statistical software IBM SPSS Statistics version 26 (IBM Corp., Armonk, NY, USA). The statistical significance level was set at *p* < 0.05.

Partial least squares structural equation modelling (PLS-SEM) was used to establish valid and reliable scales for each of the constructs and to determine the causal relationships between them. Structural equation modelling (SEM) is a multivariate data analysis technique that enables researchers to incorporate unobservable variables measured indirectly by indicator variables [36]. PLS-SEM is a variance-based SEM method that is primarily used to develop theories in exploratory research where prior knowledge is scarce, and it is a preferred method when formative constructs are included in the structural model [30,36]. A theoretical model was developed to explore the influence of objective knowledge, product attributes, and product information on subjective knowledge that acts as a mediator of fish consumption behaviour.

The analysis and interpretation of PLS modelling results were carried out for total sample and for Croatian and Italian subsample in two stages: reliability and validity of the measurement model, and evaluation of the structural model. The measurement model had four constructs with reflective indicators (objective knowledge, subjective knowledge, satisfaction with product attributes, and importance of product information) and one with formative indicators (fish consumption). Therefore, different quality criteria were applied following the relevant literature [30,36]. The evaluation of reflective constructs consisted of examining factor loadings, internal consistency reliability, convergent validity, and discriminant validity. Internal consistency reliability was measured using Cronbach’s alpha, and composite reliability with a proposed minimum of 0.7 (or 0.6 in exploratory research) and a maximum of 0.95. To establish convergent validity, the average variance extracted (AVE) was measured for all items on each construct with a threshold value of 0.5. Loadings of all indicators should be statistically significant and levels above 0.708 are recommended. Indicators between 0.4 and 0.7 should be removed only when their deletion leads to an increase in the composite reliability or leads to AVE above the threshold value, and if it does not affect the content validity of the construct. Discriminant validity was estimated using the heterotrait–monotrait ratio, the values of which should be below 0.85 for conceptually different constructs. The formative construct was evaluated based on the convergent validity, indicator collinearity, and statistical significance and relevance of the indicator weights. Convergent validity was assessed by examining the correlation of the measured construct, i.e., fish consumption, with a global single-item (redundancy analysis). The correlation should be 0.70 or higher. Variance inflation factor (VIF) was used to evaluate the collinearity of the formative indicators whose values ideally should be below 3. Indicators of the formative construct are retained if their outer weights are statistically significant, or in the case of insignificance, if their loadings are relatively high (≥0.50) and significant [30,36].

Assessment of the structural model included examining collinearity issues, significance and relevance of path coefficients, and the model’s explanatory and predictive power. Similar to the procedure of assessing formative constructs, variance inflation factors (VIF) between the constructs were checked in order to examine collinearity. The next step included assessing the coefficient of determination (R^2^) for endogenous constructs, which measures the variance and is, therefore, a measure of the model’s explanatory power. Acceptable values of R^2^ are based on the context and research discipline [36]; however, the lowest acceptable recommended level of variance explained is 10% [37]. To assess the model’s predictive accuracy, the cross-validated redundancy measure Q^2^ is used. Values should be larger than zero for each endogenous construct [30,36]. In addition to estimating direct effects, i.e., path coefficients between constructs as proposed in the structural model, the mediating effect of subjective knowledge was examined by determining indirect effects.

Furthermore, a multigroup analysis was performed to examine whether the differences between the direct and indirect paths across the countries were statistically significant. For this purpose, a nonparametric approach, partial least squares multigroup analysis (PLS-MGA), was applied. SmartPLS 3 software (SmartPLS GmbH, Oststeinbek, Germany) was used for model estimation and multigroup analysis.

## 3. Results

### 3.1. Study Participants

As seen in Table 1, gender distribution is equal in both countries, while the highest proportion of consumers from Croatia and Italy belong to the oldest age group (30.9% and 31.7%, respectively). More than half of the Croatian respondents hold a tertiary education degree (52.5%), while in Italy, the majority of respondents have secondary-school qualifications (56.4%). Regional distribution in Croatia is as follows: city of Zagreb (30%), north (14.2%), east (14.2%), centre (7.3%), south-west (13.5%) and south (20%). In Italy, the majority of study participants live in the north-west (26.5%), followed by the south (23.5%), centre (19.5%), north-east (19.3%), and islands (11.2%) (Table 1).

### 3.2. Fish Consumption

As shown in Table 2, consumers from the two countries differ significantly in terms of their fish consumption frequency (*p* < 0.001). In Italy, the majority of consumers eat white (34.2%) and fatty (40%) fish once a week. Croatians are much less-frequent fish consumers in comparison with Italians. The highest share of consumers in Croatia eat white (31.9%) and fatty (33.4%) fish 2–3 times a month.

### 3.3. Knowledge about Fish

Croatian and Italian consumers did not differ in terms of overall objective knowledge. The aggregated score for objective knowledge was 5.60 ± 0.97 and 5.55 ± 1.19 (out of 7) in Croatia and Italy, respectively, whereas no significant differences between the two countries were found based on the independent t-test (*p* = 0.356). The most commonly held knowledge was that “Fish is a source of omega-3 fatty acids”, followed by “It is recommended to eat fish at least twice a week” and “Consumption of fatty fish is important in the prevention of some chronic diseases, such as cardiovascular diseases”. Interestingly, about 60% of participants in both countries incorrectly believed that fish is a source of dietary fibre. More than one-third were not aware of the fact that sea bass and sea bream on the European market are mostly farmed. Croatian consumers showed better subjective knowledge in all questions except in those related to their ability to evaluate product quality (Table 3).

### 3.4. Product Attributes and Information

Italian consumers, in comparison with their Croatian counterparts, were more satisfied with all seafood product attributes (*p* < 0.001) (Figure 1). In both countries, consumers expressed the highest level of satisfaction with the quality and freshness of seafood on the market. Satisfaction with the price was, convincingly, the worst-rated in Italy and in Croatia (only 34% and 17% of consumers were satisfied, respectively). The highest obtained difference between the countries pertains to satisfaction with the price–quality ratio, with Italians (44%) being more satisfied than Croatian (27%) consumers (*p* < 0.001). Availability and choice are two attributes that were also significantly better-rated among Italians.

For Croatian consumers, the most important product information was shelf life, followed by country of origin and previous freezing. The two equally most important pieces of product information for Italians were shelf life and method of production (wild vs. farmed), followed by country of origin (Figure 2). Consumers from the two countries showed the greatest difference in the importance they attached to this information, relative to the importance they attached to other pieces of information.

The production method was the most important information for 76% of Italian consumers, but only for 63% of Croatian consumers (*p* < 0.001). A voluntary label related to quality was more important to Croatians (71%), while the opposite was the case with the eco-label, which was important to 64% of Italian consumers (*p* < 0.001). Nutritional value was the information about a product that consumers found to be the least important. Only 49% of respondents from Croatia and 58% of respondents from Italy considered this information as being important.

### 3.5. Measurement Model

The evaluation of reflective constructs for the total sample and for the Croatian and Italian subsample are shown in Table 4. Internal consistency reliability, measured using Cronbach’s alpha, and composite reliability exceeded the threshold value of 0.7 in all measured constructs in all samples. The measurement of average variance extracted (AVE) confirmed convergent validity, as all values reached the acceptable limit of 0.5. Almost all item loadings were above the recommended value of 0.708. One of the items was discarded (“recommended method of preparation”), while the others below the proposed limit were retained, as the internal consistency reliability and AVE were already above the threshold value without their removal, or did not substantially improve the content validity of the construct. Considering that objective knowledge was a single-item construct measured as the total sum of the correct answers of seven TRUE/FALSE items, internal consistency reliability and AVE were not appropriate measures and cannot be interpreted, since the indicator’s outer loading is fixed at 1.00. Discriminant validity was established using heterotrait–monotrait ratio values; they were between 0.042 and 0.342 in the total sample, 0.059–0.278 in the Croatian sample, and 0.034–0.467 in the Italian sample, and thus, below 0.85. Redundancy analysis confirmed the convergent validity of the formative construct, i.e., consumption of fish, while the VIF values for both indicators in all samples were below the threshold of 3 (1.530 in the total sample, 1.341 in the Croatian sample, and 1.558 in the Italian sample). Factor weights in the total sample were 0.708 and 0.404 for white and fatty fish, respectively, and were significant at a 95% confidence level. The obtained values for white fish were 0.704 and 0.726 for the Croatian and Italian samples, respectively, while the factor weight for fatty fish was 0.439 in the Croatian sample and 0.380 in the Italian sample. All factor weights in the country subsamples were statistically significant (*p* < 0.005). Figure 3 represents the measurement model of the total sample. The abbreviations of observed items displayed in rectangles are explained in Table 4 for reflective indicators, whereas the abbreviations of the formative indicators correspond to white (FC_1) and fatty (FC_2) fish consumption.

### 3.6. Structural Models and Multigroup Analysis

After the evaluation of the measurement model, the structural model was tested for both the total sample and for the Croatian and Italian subsamples. Figure 3 shows the proposed structural model applied to the total sample, while the path coefficients of the direct and indirect effects of all samples are summarised in Table 5.

In the total sample, all path coefficients were statistically significant, except for the relationship between objective and subjective knowledge. Higher satisfaction with product attributes (β = 0.197, *p* < 0.001) and a higher level of subjective knowledge (β = 0.277, *p* < 0.001) led to more frequent consumption of fish. Furthermore, satisfaction with product attributes (β = 0.282, *p* < 0.001) and the importance of product information (β = 0.161, *p* < 0.001) were positively related to subjective knowledge, while objective knowledge had a positive influence on product information (β = 0.194, *p* < 0.001) (Figure 3). In addition, objective knowledge had an indirect effect on fish consumption through product information (β = 0.009, *p* < 0.001), while product information (β = 0.045, *p* < 0.001) and product attributes (β = 0.078, *p* < 0.001) influenced fish consumption through subjective knowledge, which confirms the mediating role of subjective knowledge (Table 5). There were no collinearity issues in the model, since the maximum variance inflation factor between the constructs was 1.110 and, therefore, below the suggested threshold value of 3. Estimation of the coefficient of determination (R^2^) showed that the proposed model explains 12.5% of subjective knowledge and 15% of fish consumption behaviour (Figure 3). The cross-validated redundancy measure Q^2^ showed satisfactory predictive accuracy of the model, as values were above zero for the endogenous constructs of subjective knowledge (Q^2^ = 0.100) and fish consumption (Q^2^ = 0.115).

The quality evaluation of the structural models applied to the Croatian and Italian subsamples was also satisfactory. In the Croatian model, all path coefficients were statistically significant, whereas in the Italian model, there was no relationship between objective and subjective knowledge and no indirect effect of objective knowledge on fish consumption through subjective knowledge (Table 5). The maximum variance inflation factors were 1.076 and 1.220 in Croatia and Italy, respectively. Subjective knowledge had an R^2^ value of 13% for the Croatian dataset and 18.8% for the Italian dataset, while the R^2^ value of fish consumption was 13.4% and 18.8% for the Croatian and Italian models, respectively. The predictive relevance (Q^2^) of both models for subjective knowledge (Q^2^_CRO_ = 0.103, Q^2^_IT_ = 0.152) and fish consumption was greater than zero, thereby confirming the predictive relevance of both models (Q^2^_CRO_ = 0.095, Q^2^_IT_ = 0.127).

The results of the multigroup analysis show that the influence of product attributes on subjective knowledge was significantly greater in the Italian sample (β_CRO_ = 0.244, β_IT_ = 0.398, *p* = 0.001) while the influence of product information on subjective knowledge was significantly greater in the Croatian sample (β_CRO_ = 0.210, β_IT_ = 0.086, *p* = 0.010). Objective knowledge had a positive relationship with subjective knowledge in the Croatian sample while in the Italian sample, as already mentioned, this effect was missing (β_CRO_ = 0.089, β_IT_ = -0.052, *p* = 0.002). There were also differences in specific indirect effects, which shows that product information had a higher influence on fish consumption in the Croatian sample (β_CRO_ = 0.068, β_IT_ = 0.029, *p* = 0.026), while product attributes influenced fish consumption to a greater extent in the Italian sample (β_CRO_ = 0.079, β_IT_ = 0.134, *p* = 0.013), both through the mediator of subjective knowledge. Objective knowledge had a greater impact on fish consumption through product information (β_CRO_ = 0.014, β_IT_ = 0.005, *p* = 0.037) and through subjective knowledge (β_CRO_ = 0.029, β_IT_ = -0.018, *p* = 0.002) in the Croatian sample (Table 5).

## 4. Discussion

The results obtained in this study confirm that subjective knowledge and satisfaction with product attributes are two significant predictors of fish consumption in Croatia and Italy. The multigroup analysis reveals that there is no country-specific difference in these two constructs. Furthermore, the obtained indirect effects confirm the mediating role of subjective knowledge between product information and fish consumption, as well as between product attributes and fish consumption. Additionally, multigroup analysis shows that although satisfaction with product attributes is the strongest predictor of subjective knowledge in both countries, it has a greater effect in the Italian sample, while product information has a greater effect in the Croatian sample. Although the level of objective knowledge is high in both Croatian and Italian consumers, it positively influences subjective knowledge only in the Croatian model, while this effect is missing in the Italian model.

Italian consumers are significantly more satisfied with all product attributes in comparison with their Croatian counterparts. For participants from both countries, quality and freshness are attributes of the highest satisfaction, while the majority of consumers are not satisfied with the price and price–quality ratio. Shelf life and country of origin are very important product information for consumers in both countries while, interestingly, the production method (wild vs. farmed) is much more important for Italian than for Croatian consumers. Italian consumers are much more mindful of eco-labels, while quality labels are more important for Croatians.

The estimation of all coefficients of determination (R^2^) above the proposed limit of 10% [37] confirms the suitability of the proposed model in predicting fish consumption behaviour in both countries.

Numerous health organisations and most European Food-Based Dietary Guidelines advise the consumption of two servings of fish per week in order to ensure the provision of key nutrients. When the type of fish to be consumed is specified, most recommend that half of the consumed fish should be fatty fish [2]. However, the majority of European consumers do not follow these recommendations. The latest Eurobarometer survey shows that only one-third of Europeans eat fishery and aquaculture products at home once a week or more often [38]. Our results indicate that more than half of Italians reach the recommended frequency of fish consumption, while just around one-third of Croatians do the same.

In terms of the nutritional evaluation of fish intake, fish are divided into fatty and white fish according to their fat content and fat distribution. In fatty fish, fat is stored in fat cells throughout the body; in white fish, fat is stored in the liver and, to a lesser extent, in the abdominal cavity. White fish has a low fat content, while fatty fish contains significantly more fat. Consequently, white fish is lower in calories, but it is also rich in omega-3 fatty acids. Therefore, fatty fish is thought to provide greater health benefits than white fish due to its higher content of the desirable n-3 PUFA. In addition, white fish has a mild taste, fine texture, and is easily digested, while fatty fish has a more “meaty” taste and odour [39]. Since Italians consume significantly more fish, especially fatty fish, compared with Croatians, it can be concluded that they clearly have a better nutritional status; they especially have a higher intake of n-3 PUFA, which may have an indirect positive effect on their health, as mentioned earlier [2,3,4]. The attitudes of Italian consumers towards the two types of fish have been studied [26,33]. It was found that Italians associated white fish with positive attributes, including delicate taste, while traditional presentation forms of fatty fish were associated with negative sensory attributes [26]. Similarly, casual/non-consumers of small pelagic fish in focus studies cited taste, and poor and strong odour as reasons for low purchase rates of this type of fish [33]. However, our unpublished data did not confirm any differences between the two fish species on this parameter.

Consumer product knowledge plays an important role in decision-making and in information search processes [17]. Knowledge is one of the often-studied variables which explain consumer behaviour of buying a certain product, either directly or indirectly, in causal models examining the relationship between the constructs that formulate behaviours [13,15,40,41]. Since there is a distinction between objective and subjective knowledge [17], in our research, special attention was paid to these two aspects of knowledge. Previous studies have confirmed, similarly to ours, that consumers with a higher level of subjective knowledge eat fish more frequently [12,15,34], and that subjective knowledge is a stronger predictor of fish consumption behaviour than objective knowledge [13,15,42]. Subjective knowledge about fish also contributes to consumers’ intention to participate in various value-added-food-related activities (i.e., culinary tourism) which indirectly also contributes to higher fish consumption [43]. Among Croatian and Italian consumers, it is obvious that the level of objective knowledge is greater than that of subjective knowledge, the discrepancy being similar to the results obtained in a study among French, Spanish and Polish fish consumers [34]. Since subjective knowledge indicates an individual’s degree of self-confidence, it can be concluded that Croatian and Italian consumers are not really confident in the knowledge they possess. Even though a positive relationship between the two types of knowledge is expected and was reported in a similar study [15], the relationship between the two types of knowledge was confirmed only in the Croatian sample. However, a meta-analysis on the relationship between the measures of the two constructs reported that not all studies could provide a correlation between objective and subjective knowledge [44].

A decade ago, it was suggested that policy makers and food marketers should improve consumer objective knowledge about fish as a target outcome when communicating with consumers [16]. Those authors also pointed out the need to reconsider the labelling of fishery products with the aim of ensuring the provision of information that is of importance to consumers (i.e., nutritional value rather than fishing zone). In our study, the results pointing to a high level of objective knowledge suggest that while the objective knowledge of consumers about fish increased in the previous period, fish consumption still remains insufficient. Consequently, the activities of today’s marketers and policymakers should focus on improving the subjective knowledge of consumers about fish.

Subjective knowledge can be increased in several ways. Although our study confirmed the link between objective and subjective knowledge only in Croatia, but not in Italy, previous research confirms this connection [44]. Therefore, it can be recommended that educating the population on the benefits of fish consumption, especially children, can also improve their subjective knowledge. It is known from the literature [44] that people have a stronger sense of subjective knowledge when the information comes to them from an expert in the product category. Therefore, it is important to use credible sources of information in communication, for example, professionals and scientists. Furthermore, as emotional appeals could reinforce positive attitudes towards foods [40], they should be used in promotion to evoke positive emotions in consumers and, thus, encourage them to buy fishery products. Additionally, marketing messages can reach consumers through various channels, from advertising through traditional media (television, radio, newspapers, printed leaflets, brochures, etc.) to ads distributed through social media. In this way, the messages can reach a broad audience. In addition, one’s own usage experiences can also be a source of knowledge [45]. That is why it is important to offer consumers the opportunity to try fishery products. This can be achieved by organising tastings in shopping malls, catering facilities, educational institutions, events, etc.

Since the results of this study confirm that the consumers’ satisfaction with product attributes influences fish consumption, directly or indirectly, through subjective knowledge, it is worth examining the attributes with which consumers are the most and the least satisfied. The attributes consumers are the most satisfied with could be used by marketing professionals and fish companies in their marketing strategies, while the attributes that consumers are the least satisfied with could serve as directions for product improvement. Of the product characteristics, quality, followed by freshness, were the attributes that Croatian and Italian consumers were the most satisfied with, while they were the least satisfied with price.

Fish is often viewed as an expensive commodity. Our previous unpublished data confirm that Croatian consumers are much more price-sensitive compared with Italians and that price is a barrier to higher rates of fish consumption in both countries. Fish, however, includes a variety of products, sold at different prices [46]. Therefore, all stakeholders should seek to bolster the consumption of different varieties of fish, even cheaper ones such as small fatty fish, in order to correct that misperception. Availability, one of the attributes examined in our study, is a strong situational factor affecting the consumption of certain food, since consumers cannot buy a product that is not offered to them and they do not purchase the product if it is perceived as hard to obtain [47]. The purchase of food is often a habitual and automatised process that consumers repeat in similar conditions, so the limited availability of fish which was rated by every second participant (more in Croatia than in Italy) could be one of the obstacles to fish consumption.

The type of information related to fish in which consumers are interested, the source of information they use, and how it affects their expectation and intention to use are worth investigating [16]. An essential and almost unique source of information that consumers have about fishery products is the label [48]. The authors add that the information is relevant and demanded by consumers because fishery products are very perishable and have different origins. Indeed, shelf life and country of origin are the common most important pieces of information for Croatian and Italian consumers when buying fish products. Country of origin is the most important attribute in consumer fish choice, with domestic products being preferred over imported ones due to various reasons: safety issues, more trust in local products, ethnocentrism of consumers, etc. [49,50]. The very high importance of information on shelf life and previous freezing could be explained by the fact that most consumers are not able to evaluate the freshness of fish by examining intrinsic cues, such as the colour of the eyes and gills, and they need extrinsic cues which assist them in not making a wrong choice [46]. The fact that freshness is among the top attributes that consumers are satisfied with could imply that they have learned how to rely on information provided by food operators. However, the discrepancy between importance and satisfaction leaves room for strengthening consumer satisfaction through this attribute, especially since fresh fish products are generally much more preferred than frozen, smoked, and dried products [46,49].

Compared with Croatians, Italian consumers reported higher importance of information on the production method. As reviewed by several authors [46,49], consumers generally have a greater preference for wild fish, which originates from their perception that farmed fish are of lower quality in terms of taste, nutritional value, health, and safety. The production method has also been reported previously as one of the pieces of information in which Italian consumers are most interested, together with information on whether fish has been produced sustainably [16]. Furthermore, a recent study outlined that many consumers belonging to the “pro-aquaculture” cluster were from Italy, which confirms their more positive perception of farmed fish in comparison with the other European consumers examined [51]. In this view, and accompanied by the fact that the share of aquaculture in total fishery production in Italy (41%) is significantly higher than in Croatia (24%) [52], the Italians’ higher interest in the production method could be influenced, to some extent, by their search for farmed fish.

The importance of quality labels, as the representation of a value-added product, was more obvious among Croatian consumers, while eco-labels were more important for Italian consumers. Previous research revealed that interest in quality labels is the highest among consumers uncertain of their ability to determine seafood quality, which was also the largest segment of consumers [53]. The inability of consumers to interpret quality may be a reason for consumers placing importance on extrinsic cues, i.e., quality labels. Consumers with higher levels of subjective knowledge about fish were also more involved in fish quality and had higher knowledge about this attribute [54]. Even though Croatian and Italian consumers are satisfied with the quality of fish products, that perception needs to be further maintained, and even strengthened, among Croatians through quality labels.

As recently reviewed, consumer studies have confirmed that eco-labels enjoy good consumer recognition worldwide and are related to a higher willingness to pay [55]. Fish products sold in the Italian market could benefit more from eco-labels than those in the Croatian market. Recent research has shown that Croatian consumers are not knowledgeable about fresh organic fish. Only half of the study participants are aware of the possibility of buying fish from organic aquaculture in Croatia, and half of them are willing to pay a premium price for the fresh fish with this label [56]. Eco-labelled fish has a relatively higher price, and when considering the existing dissatisfaction of Croatian consumers with the price of fish, it is questionable whether eco-labelled products in Croatia could reach a larger group of consumers.

Consumers with higher sensitivity to price tend to have a lower preference for eco-labelled seafood [57]. However, this scenario points out the need for marketers and retailers to make a greater effort to promote such fish. This can be achieved by emphasising their high quality and locality—information that is more appreciated—while at the same, investing more time in educating consumers and communicating to them what a sustainable eco-label means [55]. Indeed, as knowledge on sustainability increases, so does the importance of the sustainability factor in seafood purchase decisions [58], while the label “Produced in own country”, together with an eco-label, functions the best as a driver of choice [59]. Furthermore, eco-labels can be beneficial in a supply chain even without a price premium, due to longer product lifespans, which can contribute to profitability [60].

The content, as well as the amount of information provided, are of importance for influencing consumers in their actions. Communication strategies should be tailored to the needs of potential users, and overlooking their demand for selective information can lead consumers to display less interest in fish products [61]. Furthermore, these authors suggest displaying less information on the front packaging and including only the relevant information that consumers are looking for, while the rest of the information, which is either mandatory or voluntary, can be made available on the back of the packaging for those who seek more. Therefore, marketers in both countries should consider highlighting some of the product information that is considered the most important in this study, such as shelf life, country of origin, and previous freezing. The method of production, together with the eco-label, should be emphasised in Italy, while in Croatia, the focus should be put on the quality label.

The strength of this study lies in its cross-cultural aspect, methodological approach, and national representative samples of fish consumers. In the last decade, partial least squares structural equation modelling has been gaining more and more attention in the research of various disciplines. Accordingly, this study proposes and tests a distinctive model on Croatian and Italian samples of fish consumers. Future studies are encouraged to apply this model in different cultural settings and to enhance it by adding other relevant constructs. The focus of this study was on fish consumption in general. Future fish consumption, however, relies on aquaculture, considering that available fish resources are limited, while at the same time, the consumption of fish is being encouraged due to its health benefits. Hence, the limitation of this study is that no special attention has been paid to the consumption of farmed fish. Future studies should examine the proposed model in the context of farmed fish consumption. Furthermore, another limitation of this study is that only fish consumption at home was considered. To obtain more detailed information on consumer habits, the consumption of fish out-of-home, as well as the consumption of other types of seafood, should be taken into account. It is also recommended that future research examine the impact of consumers’ experiences with fish consumption on satisfaction with product attributes, and the interactive relationship between satisfaction with product attributes and fish consumption. In this paper we did not analyse the influence of socioeconomic status, age, and gender on fish consumption; this should also be considered a limitation, but also a possibility that should be explored in the future. In addition, subjective and objective knowledge can be measured using other scales that may better represent these concepts.

## 5. Conclusions

National and international market developments, as well as policy regulations, should provide the general framework for changing consumer behaviour towards higher fish consumption. Efforts to improve consumers’ subjective knowledge could be an effective means to achieve this goal. Future policy interventions should focus on consumers’ self-assessed knowledge, i.e., on improving consumers’ self-awareness in evaluating all aspects of fish purchase and consumption. Country-specific differences identified in our study show that Italian consumers and the Italian market should serve as a role model for their Croatian counterparts. Since the model proposed in this research emphasises the importance of satisfaction with product attributes, it is crucial for the Croatian market to follow Italy’s example and take measures to increase satisfaction with product attributes, thus contributing to higher fish consumption.

## Figures and Tables

**Figure 1 nutrients-14-02691-f001:**
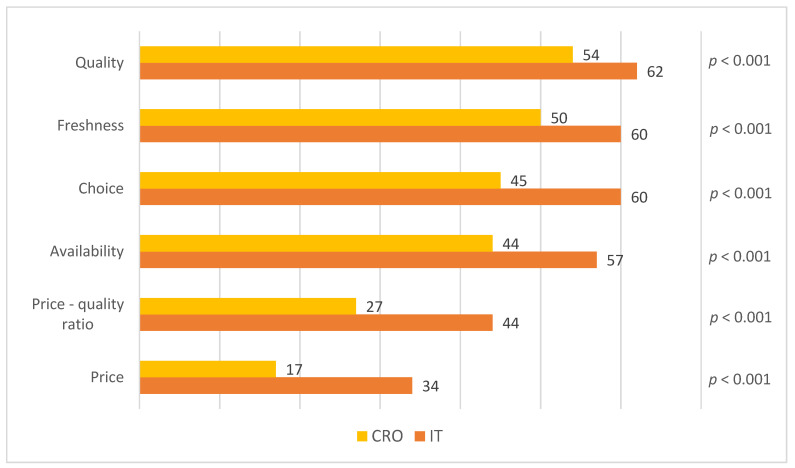
Participants’ ranking of satisfaction with product attributes in Croatia and Italy (% of participants).

**Figure 2 nutrients-14-02691-f002:**
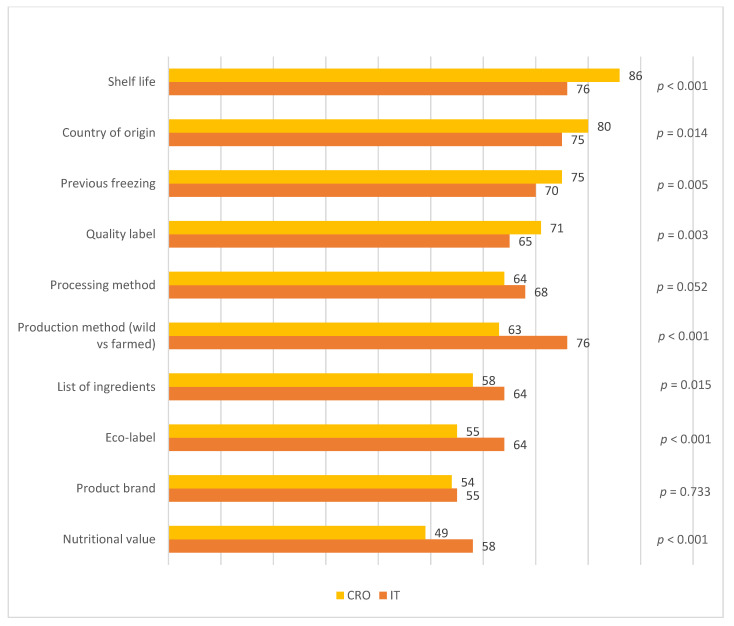
Participants’ ranking of the importance of product information in Croatia and Italy (% of participants).

**Figure 3 nutrients-14-02691-f003:**
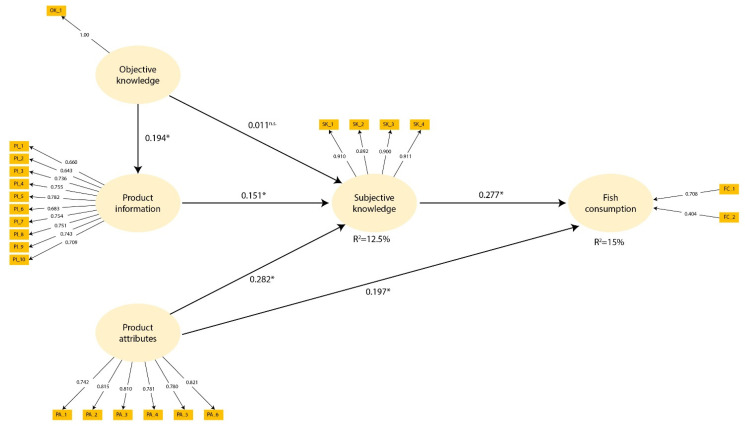
Measurement and structural model of total sample. **p* < 0.001; ^n.s^ = non significant.

**Table 1 nutrients-14-02691-t001:** Sociodemographic characteristics of study participants in Croatia and Italy.

Sociodemographic Variables		Croatia (*n* = 977)	Italy (*n* = 967)
Gender (%)	Female	49.6	49.4
	Male	50.4	50.6
Age (%)	18–30	21.5	20.5
	31–40	22.6	22.4
	41–50	25.0	25.3
	51–65	30.9	31.7
Education level (%)	Primary school or lower	1.4	5.9
	Secondary school	46.1	56.4
	Bachelor, master or higher	52.5	37.7
Average household income per month (%) *	Lower	8.3	25.2
	Middle	59.6	60.7
	Upper	14.1	11.6
	High	6.3	2.5
	N/A	11.7	0

* Croatia: lower: < HRK 5000 (< EUR 667.7); middle: HRK 5001–15,000 (EUR 667.8–2003.2); upper: HRK 15,001–20,000 (EUR 2003.3–2670.9); high > HRK 20,001 (EUR 2671.0). N/A—not applicable; HRK-Croatian currency (Kuna). * Italy: lower: < EUR 1500; middle: EUR 1501–4000; upper: EUR 4001–10,000; high: > EUR 10,000.

**Table 2 nutrients-14-02691-t002:** Frequency of consumption of white and fatty fish in study participants in Croatia and Italy.

	White Fish		Fatty Fish	
	Croatia *n* (%)	Italy *n* (%)	Croatia *n* (%)	Italy *n* (%)
1 = Once a year or less	93 (8.4%)	61 (3.1%)	82 (8.4%)	30 (3.1%)
2 = Once in 3 months	243 (24.9%)	117 (12.1%)	187 (19.1%)	91 (9.4%)
3 = 2–3 times a month	312 (31.9%)	249 (25.7%)	326 (33.4%)	217 (22.4%)
4 = Once a week	274 (28%)	331 (34.2%)	309 (31.6%)	387 (40%)
5 = 2–3 times a week	42 (4.3%)	139 (14.4%)	58 (5.9%)	185 (19.1%)
6 = 4–5 times a week	7 (0.7%)	50 (5.2%)	12 (1.2%)	40 (4.1%)
7 = Every day	6 (0.6%)	20 (2.1%)	3 (0.3%)	17 (1.8%)
*p*-value	<0.001		<0.001	
Pearson chi-squared	155.108		179.120	

**Table 3 nutrients-14-02691-t003:** Objective and subjective knowledge of study participants in Croatia and Italy.

	Correct Answer (%)		*p*-Value
Objective knowledge/Statements	Croatia	Italy	
Fish is a source of dietary fibre. (False)	41.8	39.9	0.408
Fish is a source of omega-3 fatty acids. (True)	98.2	94.3	<0.001
It is recommended to eat fish at least twice a week. (True)	96.0	91.9	<0.001
Consumption of fatty fish is important in the prevention of some chronic diseases, such as cardiovascular diseases. (True)	95.0	91.8	0.005
High maternal fish consumption during pregnancy and infant’s fish intake in the first year improves child developmental skills. (True)	72.0	79.2	<0.001
The sea bass and sea bream available in the European market are exclusively wild species. (False)	65.5	66.3	0.717
The eyes of the fish demonstrate its freshness. (True)	91.3	91.6	0.798
	Mean ± SD		*p*-value *
Aggregated score	5.60 ± 0.97	5.55 ± 1.19	0.356
	Mean ± SD		*p*-value *
Subjective knowledge/Statements	Croatia	Italy	
I consider that I know more about fish than the average person.	3.05 ± 1.08	2.89 ± 1.13	0.001
I think that I know more about fish than my friends.	3.07 ± 1.12	2.94 ± 1.16	0.013
I have a lot of knowledge about how to prepare fish.	3.14 ± 1.05	3.01 ± 1.13	0.010
I have a lot of knowledge about how to evaluate the quality of fish.	2.95 ± 1.06	2.93 ± 1.08	0.603

* Independent *t*-test.

**Table 4 nutrients-14-02691-t004:** Reflective measurement models.

Constructs	Items		Factor Loadings			Cronbach’s Alpha			Composite Reliability			Average Variance Extracted		
			Total Sample	Croatia	Italy	Total Sample	Croatia	Italy	Total Sample	Croatia	Italy	Total Sample	Croatia	Italy
Objective knowledge	OK_1		1.00 (fixed)											
Subjective knowledge	SK_1	I consider that I know more about fish than the average person	0.910	0.907	0.915	0.925	0.922	0.927	0.946	0.945	0.948	0.816	0.811	0.820
	SK_2	I think that I know more about fish than my friends	0.892	0.884	0.900									
	SK_3	I have a lot of knowledge about how to prepare fish	0.900	0.904	0.895									
	SK_4	I have a lot of knowledge about how to evaluate the quality of fish	0.911	0.906	0.912									
Product attributes	PA_1	Price	0.742	0.662	0.762	0.881	0.871	0.882	0.910	0.901	0.910	0.627	0.605	0.628
	PA_2	Quality	0.815	0.841	0.815									
	PA_3	Price–quality ratio	0.810	0.763	0.819									
	PA_4	Availability	0.781	0.768	0.784									
	PA_5	Choice	0.780	0.772	0.767									
	PA_6	Freshness	0.821	0.848	0.807									
Product information	PI_1	Shelf life	0.660	0.558	0.733	0.899	0.882	0.918	0.916	0.903	0.931	0.523	0.483	0.574
	PI_2	Nutritional value	0.643	0.600	0.692									
	PI_3	List of ingredients	0.736	0.706	0.764									
	PI_4	Country of origin	0.755	0.712	0.799									
	PI_5	Production method (wild vs. farmed)	0.782	0.773	0.816									
	PI_6	Product brand	0.683	0.660	0.713									
	PI_7	Processing method	0.754	0.717	0.795									
	PI_8	Quality label	0.751	0.747	0.759									
	PI_9	Eco-label	0.743	0.707	0.783									
	PI_10	Previous freezing	0.709	0.712	0.712									

All factor loadings were significant at *p* < 0.001.

**Table 5 nutrients-14-02691-t005:** Path coefficients of direct and indirect effects.

Path	Total Sample (n = 1944)		Croatia (n = 977)		Italy (n = 967)		Differences (Croatia vs. Italy)
Direct Effects	β	*p*-Value	β_CRO_	*p*-Value	β_IT_	*p*-Value	*p*-Value
OK → PI	0.194	<0.001	0.203	<0.001	0.188	<0.001	0.768
OK → SK	0.011	0.621	0.089	0.004	−0.052	0.076	0.002
PI → SK	0.161	<0.001	0.210	<0.001	0.086	0.016	0.010
PA → SK	0.282	<0.001	0.244	<0.001	0.398	<0.001	0.001
PA → FC	0.197	<0.001	0.104	0.001	0.131	0.004	0.575
SK → FC	0.277	<0.001	0.325	<0.001	0.336	<0.001	0.802
Indirect Effects	β	*p*-Value	β_CRO_	*p*-Value	β_IT_	*p*-Value	*p*-Value
OK → PI → SK → FC	0.009	<0.001	0.014	<0.001	0.005	0.027	0.037
OK → PI → SK	0.031	<0.001	0.043	<0.001	0.016	0.024	0.022
OK → SK → FC	0.003	0.624	0.029	0.007	−0.018	0.089	0.002
PA → SK → FC	0.078	<0.001	0.079	<0.001	0.134	<0.001	0.013
PI → SK → FC	0.045	<0.001	0.068	<0.001	0.029	0.019	0.026

OK = objective knowledge, SK = subjective knowledge, PI = product information, PA = product attributes, FC = fish consumption.

## Data Availability

The datasets generated and analysed during the current study are not publicly available, but may be available from the corresponding author upon reasonable request.

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
