# Peer review of "Fish Consumption: Influence of Knowledge, Product Information, and Satisfaction with Product Attributes"

_nutrients, 2022, doi:10.3390/nu14132691_

Round 1

Reviewer 1 Report

The aim of this manuscript was to evaluate the structure of fish consumption in two European countries- Italy and Croatia. In additin, the influence of  consumers’ knowledge,  product information, and satisfaction with product on fish consumption have salso been assessed. To generate the results, authors performed a survey by using questionnaires. All data was analyzed by using appropriate statistical tools.

Authors conclude that the national or european policy should be conducted in such a way as to encourage communities to increase consumption of fish. According to authors this can be obtained by  improving consumers' self-awareness connected with all aspects of fish purchase and consumption. Authors also pointed that Croatian market should follow the Italian one and takes appropriate measures in order to increase the satisfaction with product attributes, and inrease fish consumption among Croatian society.

The manuscript has been carefully written and it may provide some interesting information for the readers, particularly from Croatia or Italy. This may suggest, however, that this work is rather local than indended for the international readers. Notwithstanding the above, this reviewer (TR) has a few comments and questions about the manuscript:

  1. Why did the authors choose these two nations (i.e. Croats and Italians) for their study? Authors wrote “Croatia and Italy, although neighbouring Mediterranean countries sharing the Adriatic Sea, are the complete opposite in terms of fish consumption and the health status of their citizens” Is this an explanation? If so, it does not convince me at all. It rather looks like a random selection of data, because only such data were collected by the authors. In this context, there are many european countries and nations whose diets are or can be based on fish or fish products. Moreover, in fact, the authors are using the same data but in various configurations and different statistical approach.
  2. Why did the authors divide fish into two categories, i.e. white and fatty fish? Did the respondents know to which group particular fish species should be included? Did they get any schemes? Did you divide the particular fish species (broadly consumed in Europe, or in Italy and Croatia at least) into these two categories?
  3. Studies have shown, however, that chemical composition of various fish species differs significantly. Available food nutrition tables cannot be used as universal reference data for planning a balanced diet and the current recommendations for frequency and amount of fish consumption are too general. Therefore, in order to achieve better conclusions, in your study you should focus on the consumption of particular fish species and combine the survey data with chemical composition of consumed fish species (from particular countires of regions) data.
  4. Please add, as a supplementary material the entire questionnaire that was sent to the people participating in the study.
  5. Table 1: What does HRK mean? Is it Croatian currency? If so, please give (in brackets) the conversion in euro (to compare to income in Italy)
  6. Please add approproate caption of Figure 2.
  7. Authors conclude: „Future policy interventions should focus on consumers' self-assessed knowledge, i.e., on improving consumers' self-awareness in evaluating all aspects of fish purchase and consumption”. Can you discuss this issue in a broader context? (in the discussuin section). For instance, how exactly to do it? By using social media? Or on the level of basic education in school?

Author Response

We would like to express our gratitude to the Editorial Board and Reviewer 1 for their evaluation of our manuscript and useful comments. We have accepted all of the reviewer's comments and have carefully prepared our manuscript for resubmission. The reviewers' concerns are addressed below. In our revised manuscript, the changes we made at the request of reviewer 1 are highlighted in blue.

By virtue of making all of the major changes mentioned above, we have made a real effort to improve our manuscript significantly. We hope that by duly addressing the concerns of the Reviewer we have managed to eliminate all doubts about the quality of our manuscript.  In light of the position of Nutrients in academic circles, it would be of great importance for us to publish our work in your esteemed journal, so we hope that you will accept our revised paper for publication.

  1. Why did the authors choose these two nations (i.e. Croats and Italians) for their study? Authors wrote “Croatia and Italy, although neighbouring Mediterranean countries sharing the Adriatic Sea, are the complete opposite in terms of fish consumption and the health status of their citizens” Is this an explanation? If so, it does not convince me at all. It rather looks like a random selection of data, because only such data were collected by the authors. In this context, there are many european countries and nations whose diets are or can be based on fish or fish products. Moreover, in fact, the authors are using the same data but in various configurations and different statistical approach

RESPONSE: The consumer research presented in this paper is the result of work on the AdriAquaNet project- Enhancing Innovation and Sustainability in Adriatic Aquaculture, funded through ERDF, Interreg V-A Italy-Croatia 2014-2020 Program, Priority Axis- Blue innovation. EUSAIR action plan identify aquaculture as a key sector in the blue economy of Italy, Croatia and Greece, while have potentially to play a  pivotal role in the entire area. According to the aim of Project, these two countries, Croatia and Italy, have to consider each other as strong co-operators instead of competitors if they want to internationalize their markets.  One of the aim of the Project is to suggest innovative marketing system for increasing consumer awareness about importance of fish consumption. In order to propose marketing activities it is necessary to conduct market research whose part is presented in this paper. However, in our opinion, the scientific echoes of this research can contribute to a better understanding of consumer behaviour not only in these two countries but also beyond, and contribute to the choice of ways to change their behaviour in terms of increasing fish consumption as a public health interest. It is for this reason that an additional explanation of the specifics of these two countries is given.

Aware of the reviewer's opinion that we may not have argued strongly enough for the inclusion of Croatia and Italy in the research, we have clarified that the research was conducted within the project. (Ln 177-180).

  1. Why did the authors divide fish into two categories, i.e. white and fatty fish? Did the respondents know to which group particular fish species should be included? Did they get any schemes? Did you divide the particular fish species (broadly consumed in Europe, or in Italy and Croatia at least) into these two categories?

RESPONSE: We thank the reviewer for the comment. The explanation for the division of consumed fish  into white and fatty has been expanded. (Ln 184-186). Consumers knew which of the most commonly consumed fish in Croatia and Italy belonged to a particular group because this was indicated in the questionnaire (the questionnaire was included as supplementary material).

  1. Studies have shown, however, that chemical composition of various fish species differs significantly. Available food nutrition tables cannot be used as universal reference data for planning a balanced diet and the current recommendations for frequency and amount of fish consumption are too general. Therefore, in order to achieve better conclusions, in your study you should focus on the consumption of particular fish species and combine the survey data with chemical composition of consumed fish species (from particular countries of regions) data.

RESPONSE: Thanks for the comment. In accordance with the reviewer's comment, an explanation of the differences in chemical composition of white and fatty fish has been added, as well as an explanation of consumer attitudes toward the characteristics of a particular group of fish that may influence consumption (Ln 468-484).
Although it is known exactly which fish species belong to which group, it is important to keep in mind that there may be small differences in chemical composition within the same fish species depending on the area from which the fish originates, especially given the globalization of markets. For this reason, it is very difficult to accurately estimate the nutrient intake of an individual population based on the species of fish consumed, but convincing assumptions can be made based on the literature
(475-478).

  1. Please add, as a supplementary material the entire questionnaire that was sent to the people participating in the study.

RESPONSE: A portion of the questionnaire whose data were used for this paper is provided as Supplementary material and cited in the manuscript (Ln 179-180). Since this is part of a comprehensive consumer research conducted as an activity of European project, all data will be available on the project's website once the project is completed.

  1. Table1: What does HRK mean? Is it Croatian currency? If so, please give (in brackets) the conversion in euro (to compare to income in Italy)

RESPONSE: Table 1 has been modified to include all suggestions. Additionally, in accordance with Reviewer 3 comments, new categories of average household income distributions were introduced after consultation with the market research agency. The portions regarding regions have been removed from the table and commented on in the Results section, also in accordance with Reviewer 3 comments (Ln 285-288).

  1. Please add appropriate caption of Figure 2.

RESPONSE: We apologize for the mistake. Appropriate caption of  Figure 2 has been added (Ln 343-344)

  1. Authors conclude: „Future policy interventions should focus on consumers' self-assessed knowledge, i.e., on improving consumers' self-awareness in evaluating all aspects of fish purchase and consumption”. Can you discuss this issue in a broader context? (in the discussion section). For instance, how exactly to do it? By using social media? Or on the level of basic education in school?

RESPONSE: Thanks to the reviewer for the comment. The discussion of the paper is expanded to include ways of expanding knowledge, in line with the reviewer's comments (Ln 525-540).

Reviewer 2 Report

The article “Fish consumption: influence of knowledge, product information, and satisfaction with product attributes” is very well written and designed. The methodology is described in details. The discussion is interesting and raises these issues that require more extensive exploration in the context of the subject matter under consideration. Below I present my remarks:

1.                  The objective knowledge was the single-item construct in PLS-SEM. I could not find whether was formally explained  the method of construction of one reflective indicator based on seven TRUE/FALSE  items comprising on objective knowledge questionnaires. Was it just the score equal to total sum of all correct answers? The remaining items of other constructs were not dichotomised but measured on Likert scales and were incorporated as separate indicators. Would be possible in PLS-SEM model to incorporate 7 separate TRUE/FALSE items as reflective indicators of objective knowledge?

2.                  In Figure 3 “Product information” was put twice. This mistake cause difficulty with following results and is misleading The bottom one should be corrected and replaced by “Product attributes”.

3.                  Please unify the notation Q2, Q2, Q2 (the cross-validated redundancy measure). The same with R2.

4.                  The Authors built a theoretical model which structure assumed that subjective knowledge had a mediating role between fish consumption and other examined variables. Additionaly Product information mediate in the association between Objective and subjective knowledge. Could you give in Introduction or Methodology the literature which can support some assumed theoretical framework? PLS-SEM is a confirmatory tool rather than an exploratory tool. In applications, a clear objective of the study, and some basic background about the key structure of the model should be known – either from the subject knowledge or from preliminary data analysis.

5.                  Similarly in some places e.g. in “regular fish consumption could address the burden of non-communicable diseases associated with overweight or obesity, but also micronutrient deficiency, cardiovascular disease mortality, heart failure, stroke, depression and mental illnesses [7, 8].” more literature would be beneficial (here e.g. DOI: 10.1016/j.jad.2016.08.011)

6.                  The Authors wrote that “Convergent validity was assessed by examining the correlation of the measured construct, i.e., fish consumption, with a global single-item (redundancy analysis).” And next that “Redundancy analysis confirmed the convergent validity of the formative construct, i.e., consumption of fish (data not shown),”. Is there information anywhere how  was designed global single-item used in redundancy analysis in questonnaire about fish consumption? I found the information: “Fish consumption was measured as the self-reported frequency of consumption of two generic categories of fish: white fish and fatty fish…. Levels of consumption frequency were encoded in seven rank ordinal categories: once a year or less, once in 3 months, 2-3 times a month, once a week, 2-3 times a week,  4-5 times a week, every day.” However I dod not find the information how did look a global single-item used in redundancy analysis. Was it also seven leveled Likerd scale – but without division to type of fish (white vs fatty).

7.                  As a rule of thumb, Q² values higher than 0, 0.25, and 0.5 depict small, medium, and large predictive relevance of the PLS-path model. Could you put more information about this value (besides that is positive).

8.                  N/A for a sake of completeness can be explained under the Table 1. And maybe to unify the style of presentation, average household income per month can be included in the same row for both countries - different cut-off points for a given five categories (e.g. high/…./low) can be assigned under the table for easier readability. Then regions can be described just only in the text – than always all cells would be filled in a table.

9.                  “The aggregated score for objective knowledge was 5.6” – is this a mean value? What about SD? P-value presented was based on t-Student test in Table 3?

10.              In Figure 1. Participants´ ranking of satisfaction with product attributes in Croatia and Italy % of participants is presented. But the level of satisfaction with product attributes was measured on a 5-point Likert scale. What statistic is presented as percentage? Is this % of persons who ranked the satisfaction with the quality highest?

11.              The importance of product information -the same question as in the 9 point above

Were Likert questions dichotomised somehow to obtained “%”, e.g is “the importance” category  = 5 very important” a cut-off point?

12.              Above paragraph 3.5., the signature under Figure 2 is cut.

13.              “The nutritional value of the product in Croatia and Italy was rated low (10th place out of 10 and 9th place out of 10, respectively” – This is not clear how was evaluated the place 9 and 10th – was it based on the mean on the Likert scale or just percentage of highest answers (5=very important)?

Author Response

We would like to express our gratitude to the Editorial Board and Reviewer 2 for their evaluation of our manuscript and useful comments. We have accepted all of the reviewer's comments and have carefully prepared our manuscript for resubmission. The reviewers' concerns are addressed below. In our revised manuscript, the changes we made at the request of Reviewer 2 are highlighted in red.

By virtue of making all of the major changes mentioned above, we have made a real effort to improve our manuscript significantly. We hope that by duly addressing the concerns of the Reviewer we have managed to eliminate all doubts about the quality of our manuscript.  In light of the position of Nutrients in academic circles, it would be of great importance for us to publish our work in your esteemed journal, so we hope that you will accept our revised paper for publication.

This study explored the factors influencing fish consumption in Italian and Croatian people. The Investigated factors were objective knowledge, subjective knowledge, production attributes, and production information.

  1. However, the concept of the modeling is not clear. “Product information”, and “satisfaction with product attributes” (Line 107) were not introduced before Line 103

RESPONSE:  We thank the reviewer for pointing out this drawback. We have added to the introduction a description of the importance of product information and satisfaction with product attributes, and their role in proposed model (Ln 92-121). We hope that we have eliminated the weak points and improved the quality of the work.

  1. Furthermore, “satisfaction” is to fulfill one’s expectations, needs, or demands. It indicates that one has their expectations from their knowledge, and they experience, here, buying fish. So, satisfaction is not an antecedent variable to fish consumption but a consequence variable from fish consumption.

RESPONSE:  Thank you for your observation. We agree that according to consumer behaviour studies, satisfaction is a consequence of consumption or the result of meeting expectations. However, it is well known that satisfaction with a product or service is a determinant of repeat purchases. We have argued this in the text of the article, citing relevant sources (Ln 103-121).

  1. Product information (Line 107), and importance of production information (Line 139) are unclear. The phrases may be the same to “the perceived importance of this information”, which appeared in Line 278 for the first time. This is precisely ones’ belief, an acceptance that something exists or is true. The definition should be clarified.

RESPONSE: Thanks for the helpful comment that the terminology is not harmonized.  In order to ensure clarity, the terminology is harmonized  throughout the paper and we used the term "importance of product information" which is in line with the question in the survey.

  1. It is unknown whether quizzes in measuring objective knowledge are appropriate. They may be unsupported superstition, or vulgar believes. “High maternal fish consumption during pregnancy and infant’s fish intake in the first year improves child developmental skill.” Over consumption of fish involving mercury has an adverse effect. “It is recommended to eat fish at least twice a week.” “The eyes of the fish demonstrate its freshness.” It depends on kinds of fish. Moreover, four of quizzes were correctly answered >90%. These variables could not differentiate the sample population.

RESPONSE: We agree with the reviewer's comment that they may be too simple and cover basic knowledge, which is confirmed by the high percentage of correct answers, but to our knowledge there is no other questionnaire yet that is scientifically valuable and cited. However, the questions we used to assess objective knowledge are widely cited in the literature (Almeida et al., 2015; Pieniak et al., 2010; Pieniak et al., 2013), so this was our reason for including them.

We are aware of the fact that the reviewer emphasizes that objective knowledge is great, which should be due to increased awareness of healthy nutrition in general and particularly fish consumption, so we have emphasized the importance of increasing subjective knowledge (Ln 525-540).

  1. The concept should be explained in the Introduction.

RESPONSE: Thank you for your useful comment.  At the reviewer's suggestion, we have tried to expand the introduction considerably so that all the elements of the model and their interrelationships are well explained (Ln 92-121).  We hope that we have eliminated the weak points and improved the quality of the work.

  1. The authors described the different recommendations between Italy and Croatia. It is not defined.

RESPONSE: We agree with the reviewer's suggestion that it is not necessary to highlight the recommendations for Croatia and Italy separately, what could be confusing. It is sufficient to list the EFSA recommendations which are widely accepted. In this context, the sentences have been deleted for better understanding (Ln 464-467).

  1. Second, the sample population is inappropriate. 1) “Participants were a nationally representative sample of consumers in Croatia and Italy who are responsible for food purchasing within the household.” (Line 128) Maybe females are dominant. But the authors conducted “post-stratification, RIM weighting by gender, age and region according to the latest estimates of the State Statistical Offices of Croatia and Italy.” (Line 125-127) This maneuver averaged the sample between male and females. It did not a representative population responsible for food purchasing. First sampling may be a cluster sampling based on households (?).

RESPONSE: We thank the reviewer for pointing out that it can be concluded from our description that the sample is not a nationally representative sample of individuals responsible for household food purchases. It is logical to expect that women will be more heavily represented in this sample. The reviewer correctly concluded that this was due to the post-stratification procedure and  RIM weighting.
Since the research was conducted by a market research  agency, we asked them  for clarification.
Following their response, we regret to apologize as we found that we had misinterpreted the sample selection methodology and on this occasion we kindly ask the reviewer to accept the new clarification that is consistent with the actual research conducted in the field.
To conlude, our participants were a nationally representative sample of people  who are responsible for food purchasing within the household in Croatia and Italy. The detailed description  of data collection provided by market research agency IPSOS  we  have included in our revised manuscript (Ln 160-171) but also provide clarification for reviewer as follows:

In this project, however, RIM weighting was not done. Since the purpose of weighting is to harmonize the achieved sample in line with the ideal proportions obtained from the last census or with the data known about the population, and the agency did not have data on consumers of fishery products, it was not possible to perform weighting data.
We apologize for this misinterpretation. This sentence will be deleted (Ln 149-153).
In addition, the explanation for why there were not more women in the sample (one might  assume that they are mainly responsible for grocery shopping) lies in the following methodology:
The agency had drawn a stratified random sampling and a proportional quota sampling at the beginning, and then screened out people who are not in charge of buying food for the household.
There were 4 possible responses to the question:
Are you the person most often responsible for groceries in your household, or is it another household member?
1. Me personally 2. Me and another household member equally 3. Another household member 4. Can not decide
Only respondents who answered: Me personally,  and Me and another household member in equal shares participated in the questionnaire.
In terms of gender, there were more female respondents who reported being the main household shopper (Me personally), 62% in Croatia and 58% in Italy. However, when the agency looks at both categories (Me personally + Me and another household member equally), the distribution between the two genders was  equal. According to the agency, this is something it regularly observes in all studies.

  1. In addition, it is not clear why samples from Italy and Croatia were coequally combined for the main analysis. As the first step, separate analysis should be carried out, and the next step is to compare two models.

RESPONSE: In accordance with the requirements of the reviewer, the presentation of the results of the analyses performed was revised and expanded. In addition to the measurement models for the entire sample, measurement models specifically for Italy and Croatia are presented (revised Table 4) and described (Ln 232-233; 347-368).  The results of the analysis of structural models for Italy and Croatia are presented, followed by a comparison of the models performed using multigroup analysis.

  1. In Fig. 3, observed variables (indicator variables) should be depicted in rectangles with coefficients. There are two unobservable variables “product information.” Triple asterisks are not needed.

RESPONSE: We apologize for type-writing error. We have presented a completely new Figure 3, in which all elements have been included according to the reviewer's suggestions.

  1. The Introduction and Discussion are too long. Several paragraphs are not needed. For example, the first and fifth paragraphs of the Introduction should be deleted.

RESPONSE: The first and fifth paragraphs of the introduction have been deleted on the reviewer's instructions. Some parts of the discussion were deleted, as suggested but some parts are  expanded based on the comments of the other two reviewers (Ln 35-46; Ln 70-74).

Reviewer 3 Report

This study explored the factors influencing fish consumption in Italian and Croatian people. The Investigated factors were objective knowledge, subjective knowledge, production attributes, and production information. However, the concept of the modeling is not clear. “Product information”, and “satisfaction with product attributes” (Line 107) were not introduced before Line 103. Furthermore, “satisfaction” is to fulfill one’s expectations, needs, or demands. It indicates that one has their expectations from their knowledge, and they experience, here, buying fish. So, satisfaction is not an antecedent variable to fish consumption but a consequence variable from fish consumption. Product information (Line 107), and importance of production information (Line 139) are unclear. The phrases may be the same to “the perceived importance of this information”, which appeared in Line 278 for the first time. This is precisely ones’ belief, an acceptance that something exists or is true. The definition should be clarified. It is unknown whether quizzes in measuring objective knowledge are appropriate. They may be unsupported superstition, or vulgar believes. “High maternal fish consumption during pregnancy and infant’s fish intake in the first year improves child developmental skill.” Over consumption of fish involving mercury has an adverse effect. “It is recommended to eat fish at least twice a week.” The authors described the different recommendations between Italy and Croatia. It is not defined. “The eyes of the fish demonstrate its freshness.” It depends on kinds of fish. Moreover, four of seven quizzes were correctly answered >90%. These variables could not differentiate the sample population. The concept should be explained in the Introduction.

Second, the sample population is inappropriate. 1) “Participants were a nationally representative sample of consumers in Croatia and Italy who are responsible for food purchasing within the household.” (Line 128) Maybe females are dominant. But the authors conducted “post-stratification, RIM weighting by gender, age and region according to the latest estimates of the State Statistical Offices of Croatia and Italy.” (Line 125-127) This maneuver averaged the sample between male and females. It did not a representative population responsible for food purchasing. First sampling may be a cluster sampling based on households (?). In addition, it is not clear why samples from Italy and Croatia were coequally combined for the main analysis. As the first step, separate analysis should be carried out, and the next step is to compare two models.

In Fig. 3, observed variables (indicator variables) should be depicted in rectangles with coefficients. There are two unobservable variables “product information.” Triple asterisks are not needed.

The Introduction and Discussion are too long. Several paragraphs are not needed. For example, the first and fifth paragraphs of the Introduction should be deleted.

Author Response

We would like to express our gratitude to the Editorial Board and Reviewer 3 for their evaluation of our manuscript and useful comments. We have accepted all of the reviewer's comments and have carefully prepared our manuscript for resubmission. The reviewers' concerns are addressed below. In our revised manuscript, the changes we made at the request of Reviewer 3 are highlighted in green.  
By virtue of making all of the major changes mentioned above, we have made a real effort to improve our manuscript significantly. We hope that by duly addressing the concerns of the Reviewer we have managed to eliminate all doubts about the quality of our manuscript In light of the position of Nutrients in academic circles, it would be of great importance for us to publish our work in your esteemed journal, so we hope that you will accept our revised paper for publication.

The article “Fish consumption: influence of knowledge, product information, and satisfaction with product attributes” is very well written and designed. The methodology is described in details. The discussion is interesting and raises these issues that require more extensive exploration in the context of the subject matter under consideration.

Below I present my remarks:

  1. The objective knowledge was the single-item construct in PLS-SEM. I could not find whether was formally explained the method of construction of one reflective indicator based on seven TRUE/FALSE items comprising on objective knowledge questionnaires. Was it just the score equal to total sum of all correct answers? The remaining items of other constructs were not dichotomised but measured on Likert scales and were incorporated as separate indicators. Would be possible in PLS-SEM model to incorporate 7 separate TRUE/FALSE items as reflective indicators of objective knowledge?

RESPONSE: We followed a procedure based on previous research (Pieniak et al., 2010; Pieniak et al., 2010a; Pieniak et al., 2013) in which objective knowledge was assessed as the sum of correct answers giving one variable ultimately which then was entered as single-item construct in SEM analysis.

In the Methodology section, lines 137-138, there is an explanation provided: "The number of correct answers was summed for each participant, giving an aggregated score of objective knowledge on a scale of 0-7 [19]." In addition, in the results section the sentence in lines 356-359  was reformulated to ensure more clarity” Considering that objective knowledge was a single-item construct measured as a total sum of correct answers of seven TRUE/FALSE items internal consistency reliability and AVE were not appropriate measures and cannot be interpreted since the indicator's outer loading is fixed at 1.00”

  1. In Figure 3 “Product information” was put twice. This mistake cause difficulty with following results and is misleading The bottom one should be corrected and replaced by “Product attributes”.

RESPONSE: Thank you for your useful comment. We apologize for type-writing error. We have presented a completely new Figure 3, in which all elements have been included according to the reviewer's suggestions.

  1. Please unify the notation Q2, Q2, Q2 (the cross-validated redundancy measure). The same with

RESPONSE:  Thank you for your comment. We have carefully checked through the manuscript and make requested  corrections.

  1. The Authors built a theoretical model which structure assumed that subjective knowledge had a mediating role between fish consumption and other examined variables. Additionally Product information mediate in the association between Objective and subjective knowledge. Could you give in Introduction or Methodology the literature which can support some assumed theoretical framework? PLS-SEM is a confirmatory tool rather than an exploratory tool. In applications, a clear objective of the study, and some basic background about the key structure of the model should be known – either from the subject knowledge or from preliminary data analysis.

RESPONSE:  At the reviewer's suggestion, we have tried to expand the introduction considerably so that all the elements of the model and their interrelationships are well explained (92-121). We hope that we have eliminated the weak points and improved the quality of the work.

  1. Similarly in some places e.g. in “regular fish consumption could address the burden of non-communicable diseases associated with overweight or obesity, but also micronutrient deficiency, cardiovascular disease mortality, heart failure, stroke, depression and mental illnesses [7, 8].” more literature would be beneficial (here e.g. DOI: 10.1016/j.jad.2016.08.011)

RESPONSE: The recommended reference is very useful and we have included it in the paper, according to reviewer´s suggestion (Ln 54-55).

  1. The Authors wrote that “Convergent validity was assessed by examining the correlation of the measured construct, i.e., fish consumption, with a global single-item (redundancy analysis).” And next that “Redundancy analysis confirmed the convergent validity of the formative construct, i.e., consumption of fish (data not shown),”. Is there information anywhere how was designed global single-item used in redundancy analysis in questionnaire about fish consumption? I found the information: “Fish consumption was measured as the self-reported frequency of consumption of two generic categories of fish: white fish and fatty fish…. Levels of consumption frequency were encoded in seven rank ordinal categories: once a year or less, once in 3 months, 2-3 times a month, once a week, 2-3 times a week, 4-5 times a week, every day.” However I dod not find the information how did look a global single-item used in redundancy analysis. Was it also seven level Likerd scale – but without division to type of fish (white vs fatty).

RESPONSE: We agree with the reviewer that we have not provided a sufficiently detailed description and thank him for the suggestion. We have made an addition according to the instructions (Ln 189-190)

  1. As a rule of thumb, Q² values higher than 0, 0.25, and 0.5 depict small, medium, and large predictive relevance of the PLS-path model. Could you put more information about this value (besides that is positive).

RESPONSE: Thank you for your comment. The Q2 values have been added (Ln 399-400).

  1. N/A for a sake of completeness can be explained under the Table 1. And maybe to unify the style of presentation, average household income per month can be included in the same row for both countries - different cut-off points for a given five categories (e.g. high/…./low) can be assigned under the table for easier readability. Then regions can be described just only in the text – than always all cells would be filled in a table.

RESPONSE: Table 1 was revised in accordance with the reviewer's instructions. After consultation with the market research agency, a new distribution of average household income (in 4 categories for each country; with cut-off points) has been introduced. The omitted parts regarding regions have been removed from the table and commented in the Results section (Ln 280-283).

  1. The aggregated score for objective knowledge was 5.6” – is this a mean value? What about SD? P-value presented was based on t-Student test in Table 3?

RESPONSE: We thank you for the suggestion and know that clarification is needed. We have included the explanation in the text  (Ln 303-305).

  1. In Figure 1. Participants´ ranking of satisfaction with product attributes in Croatia and Italy % of participants is presented. But the level of satisfaction with product attributes was measured on a 5-point Likert scale. What statistic is presented as percentage? Is this % of persons who ranked the satisfaction with the quality highest?

RESPONSE: In the Methodology section additional explanation is provided (Ln 203-205).

  1. The importance of product information -the same question as in the 9 point above Were Likert questions dichotomised somehow to obtained “%”, e.g is “the importance” category = 5 very important” a cut-off point?

RESPONSE:  In the Methodology section additional explanation is provided (Ln 212-213)

  1. Above paragraph 3.5., the signature under Figure 2 is cut.

RESPONSE: Appropriate caption of  Figure 2 has been added

  1. “The nutritional value of the product in Croatia and Italy was rated low (10th place out of 10 and 9th place out of 10, respectively” – This is not clear how was evaluated the place 9 and 10th – was it based on the mean on the Likert scale or just percentage of highest answers (5=very important)?

RESPONSE: We agree with the reviewer's comment that this way of presenting the results is unclear. Ninth and tenth place were determined based on the numerical value of the percentage of respondents who rated it as important. To make this clear, and easier for reader to follow,  the sentence was rephrased (Ln 339-341).

Round 2

Reviewer 1 Report

Most of my questions have been addressed and the manuscript has been revised accordingly.

Author Response

The authors are indebted to our esteemed Reviewer for helpful suggestions and comments.

Reviewer 2 Report

The Authors made a huge effort to improve the quality of the manuscript. They explained very detailed all reviewers questions and hesitations, and corrected or justified many methodological concepts. For me, everything now is clear.

Author Response

(The authors gave the same response as above.)

Reviewer 3 Report

The comments responded to by the authors are not mine.

The authors explained satisfaction in the Introduction. “Satisfaction with product attributes may play an important role in future fish consumption. Customer satisfaction is an individual's perception of the performance of the product or service relative to their expectations [20]. However, consumer experiences and post-purchase evaluations play a major role in future purchase decisions [21]. Previous research has confirmed that consumer satisfaction influences intention to repurchase food [22].” Purchase experience and post-purchase evaluations form customer satisfaction, and satisfaction leads future purchase. This interactive relationship suggests a bidirectional effects between satisfaction (product attributes) and fish consumption. In Figure 3, a bidirectional arrow should be used. So, the relationship between subjective knowledge and product attributes should be inversed. Together, the discussion should be revised.

In Figure 3, the coefficient between OK_1 and objective knowledge is 1.00. However, it is reasonable that there may be unmeasured, unexpected, or unconsidered objective knowledge. Seven items the authors used cannot comprehend subjective knowledge. For example, school exams are not perfect tools. This flaw is caused by that the authors did not consider error factors. Other coefficients between unobservable factors and observed items are unexpectedly high. Furthermore, socioeconomic status, age, and gender were not involved in the analysis, error factors cannot be ignored. Errors should be considered in the analysis.

In Figures 3, what items in the questionnaire (Tables 2, 3, and Figures 1, and 2) correspond to observed items in rectangles?

Editing of English language and style used in the added and corrected highlight parts is required.

Author Response

We would like to express our gratitude to the Editorial Board and Reviewer 3 for their evaluation of our manuscript and useful comments.

We have carefully prepared our manuscript for resubmission. The reviewers' concerns are addressed below. In our revised manuscript, the changes we made at the request of reviewer 3 are marked as track changes.

  1. The authors explained satisfaction in the Introduction. “Satisfaction with product attributes may play an important role in future fish consumption. Customer satisfaction is an individual's perception of the performance of the product or service relative to their expectations [20]. However, consumer experiences and post-purchase evaluations play a major role in future purchase decisions [21]. Previous research has confirmed that consumer satisfaction influences intention to repurchase food [22].” Purchase experience and post-purchase evaluations form customer satisfaction, and satisfaction leads future purchase. This interactive relationship suggests a bidirectional effects between satisfaction (product attributes) and fish consumption. In Figure 3, a bidirectional arrow should be used. So, the relationship between subjective knowledge and product attributes should be inversed. Together, the discussion should be revised.

RESPONSE: We thank the reviewer for suggesting an interactive relationship between satisfaction with product attributes and fish consumption and that the interactive relationship between the constructs should be explored. Unfortunately, such a relationship cannot be modelled using PLS-SEM according to Hair et al. (Hair, J.F.; Hult, G.T.M.; Ringle, C.M.; Sarstedt, M. A Primer on Partial Least Squares Structural Equation Modelling (PLS-SEM), SAGE Publications: Thousand Oaks, CA, USA, 2014, pp. 15-17). PLS-SEM can only work with recursive models without causal loops and circular relationships. Thus, as a recommendation for future work, we proposed to examine the impact of consumers' experiences with fish consumption on satisfaction with product attributes ( Line: 707-710).

  1. In Figure 3, the coefficient between OK_1 and objective knowledge is 1.00. However, it is reasonable that there may be unmeasured, unexpected, or unconsidered objective knowledge. Seven items the authors used cannot comprehend subjective knowledge. For example, school exams are not perfect tools. This flaw is caused by that the authors did not consider error factors. Other coefficients between unobservable factors and observed items are unexpectedly high. Furthermore, socioeconomic status, age, and gender were not involved in the analysis, error factors cannot be ignored. Errors should be considered in the analysis.

RESPONSE: Subjective and objective knowledge were examined using scales previously used in the literature. ("Consumers' level of objective knowledge was measured with seven statements that were either true or false, based on previous studies [12, 16, 34]. Subjective knowledge was measured with four statements on a 5-point Likert scale ranging from 1 (strongly disagree) to 5 (strongly agree), based on a previous study [35].) We agree that the existing approach may not encompass all possible variables and that the use of other approaches and scales to measure knowledge is possible. Therefore, we mentioned this in the limitations of the paper and suggested using other measures to determine subjective and objective knowledge in future studies. The same is true for socioeconomic status, age, and gender, which is a shortcoming of this paper. Therefore, we suggest that the impact of these variables should be investigated in future studies. (Lines: 710-714). We thank the reviewer for pointing out the consideration of error factors. We would like to explain that in the analysis we conducted, all reflective indicators as well as endogenous constructs are associated with an error term, whereas the objective of PLS-SEM is to use the available data in order to estimate path relationships while minimizing error terms. Although we are aware of the possibility of conducting additional analyses, unfortunately we do not have the capacity to do so at this time. The project under which the research was funded has ended, and we no longer have a license. We will consider these valuable proposals in our future work. . We hope you accept our apologies.

  1. In Figures 3, what items in the questionnaire (Tables 2, 3, and Figures 1, and 2) correspond to observed items in rectangles?

RESPONSE: We thank the reviewer for the comment. Since Figure 3 represents Measurement and structural model, the explanations of the observed items in the rectangles are included in Table 4. In Table 4, a new column is added (to the left of the construct) where all the observed items in the rectangles are linked to the items in the questionnaire. In addition, explanations of FC 1 and FC2 are provided in lines 398-401.

  1. Editing of English language and style used in the added and corrected highlight parts is required.

RESPONSE: We thank the reviewer for the comment. All added parts are again carefully checked and corrected by a native English speaker. These parts are deleted by the track-changes option and new parts are inserted (Lines: 42-46, 90-96, 118-137, 151-161, 198-200, 227-229, 238-239, 314-318, 363-365, 372-374, 403-406, 513-531, 580- 597).

By virtue of making all of the changes mentioned above, we have made a real effort to improve our manuscript significantly. We hope that by duly addressing the concerns of the Reviewer we have managed to eliminate all doubts about the quality of our manuscript.  In light of the position of Nutrients in academic circles, it would be of great importance for us to publish our work in your esteemed journal, so we hope that you will accept our revised paper for publication.
